# Heterogeneity in Signaling Pathway Activity within Primary and between Primary and Metastatic Breast Cancer

**DOI:** 10.3390/cancers13061345

**Published:** 2021-03-16

**Authors:** Márcia A. Inda, Paul van Swinderen, Anne van Brussel, Cathy B. Moelans, Wim Verhaegh, Hans van Zon, Eveline den Biezen, Jan Willem Bikker, Paul J. van Diest, Anja van de Stolpe

**Affiliations:** 1Precision Diagnostics Department, Philips Research, 5656 AE Eindhoven, The Netherlands; marciainda@hotmail.com (M.A.I.); paul@vanswinderen.com (P.v.S.); hvanzon@gmail.com (H.v.Z.); 2Philips Molecular Pathway Diagnostics, 5656 AE Eindhoven, The Netherlands; anne.brussel@philips.com (A.v.B.); eveline.den.biezen@philips.com (E.d.B.); anja.van.de.stolpe@philips.com (A.v.d.S.); 3Department of Pathology, University Medical Center Utrecht, 3508 GA Utrecht, The Netherlands; cmoelans@umcutrecht.nl (C.B.M.); p.j.vandiest@umcutrecht.nl (P.J.v.D.); 4CQM, Consultants in Quantitative Methods, 5616 RM Eindhoven, The Netherlands; janwillem.bikker@cqm.nl

**Keywords:** tumor heterogeneity, signaling pathway activity, mRNA analysis, primary tumor, metastasis, breast cancer

## Abstract

**Simple Summary:**

Personalized breast cancer treatment with targeted therapy (e.g., tamoxifen or PI3K inhibitors) requires identification of responder patients. Phenotypical heterogeneity within the primary, and between primary tumor and metastases, may however interfere with response to therapy, if based on a single primary tumor biopsy. In this study, we investigated this heterogeneity, using novel assays to measure activity of tumor-driving signal transduction pathways, e.g., estrogen receptor and PI3K pathways, in multiple samples distributed across the tumor and in metastases. Within the primary tumor, heterogeneity was dominant at microscale (biopsy block) and not at macroscale (across tumor), suggesting that a single biopsy is generally representative for the whole primary tumor. The differences found between pathway activity in primary tumor and metastases suggests that it is recommendable to analyze pathway activity in metastatic samples for treatment selection for late stage patients.

**Abstract:**

Targeted therapy aims to block tumor-driving signaling pathways and is generally based on analysis of one primary tumor (PT) biopsy. Tumor heterogeneity within PT and between PT and metastatic breast lesions may, however, impact the effect of a chosen therapy. Whereas studies are available that investigate genetic heterogeneity, we present results on phenotypic heterogeneity by analyzing the variation in the functional activity of signal transduction pathways, using an earlier developed platform to measure such activity from mRNA measurements of pathways’ direct target genes. Statistical analysis comparing macro-scale variation in pathway activity on up to five spatially distributed PT tissue blocks (*n* = 35), to micro-scale variation in activity on four adjacent samples of a single PT tissue block (*n* = 17), showed that macro-scale variation was not larger than micro-scale variation, except possibly for the PI3K pathway. Simulations using a “checkerboard clone-size” model showed that multiple small clones could explain the higher micro-scale variation in activity found for the TGFβ and Hedgehog pathways, and that intermediate/large clones could explain the possibly higher macro-scale variation of the PI3K pathway. While within PT, pathway activities presented a highly positive correlation, correlations weakened between PT and lymph node metastases (*n* = 9), becoming even worse for PT and distant metastases (*n* = 9), including a negative correlation for the ER pathway. While analysis of multiple sub-samples of a single biopsy may be sufficient to predict PT response to targeted therapies, metastatic breast cancer treatment prediction requires analysis of metastatic biopsies. Our findings on phenotypic intra-tumor heterogeneity are compatible with emerging ideas on a Big Bang type of cancer evolution in which macro-scale heterogeneity appears not dominant.

## 1. Introduction

Cancer can be described in terms of abnormal functioning of one or more signal transduction pathways that control major cellular functions, e.g., cell division, differentiation, migration, and metabolism. Around 10–15 signal transduction pathways can drive growth and metastasis of breast cancer [1,2]. In cancer, they can be activated by receptor ligands or specific DNA mutations [3,4,5,6,7,8,9,10,11]. Signaling pathways are in principle clinically actionable, since their activity can be modified by certain drugs or other treatments. Treatment with targeted drugs is increasingly used in breast cancer, aiming to block the tumor-driving pathways in a (neo)adjuvant or metastatic setting [12]. However, predicting the effect of targeted therapy has generally proven to be very difficult [13]. One reason is that cancer genome mutation analysis does not sufficiently predict which signaling pathways are active in an individual tumor [13,14]. Furthermore, heterogeneity in signaling pathway activity may be present within a tumor, while targeted drug choice is usually based on analysis of a single primary tumor (PT) biopsy. Similar problems exist when choosing targeted therapy for treatment of patients with metastatic disease, where therapy choice is generally based on analysis of a tissue sample from the PT, although marked genotypic and phenotypic differences between PT and distant site (DS) metastases have been described [15,16]. Unfortunately, limited knowledge is available on heterogeneity in signaling pathway activity within a PT and between PT and lymph node (LN) and DS metastases [17,18]. One reason has been the lack of reliable assays to measure signaling pathway activity in formalin-fixed paraffin-embedded (FFPE) tissue samples used in routine diagnostic settings.

A novel analysis method was described before to quantify signaling pathway activity in cancer. Based on Bayesian models, the method infers a pathway activity score from transcription factor-specific target gene mRNA levels [19,20,21,22]. While originally developed for use on fresh frozen tissue samples, this method was recently adapted for use on FFPE material for a number of signaling pathways, i.e., androgen receptor (AR), estrogen receptor (ER), PI3K-FOXO, Hedgehog (HH), TGFβ and Wnt pathways [23,24,25]; see Appendix A). Using this approach, we analyzed heterogeneity in signaling pathway activity within breast PT and between PT and LN and DS metastases.

## 2. Materials and Methods

### 2.1. Study Design

The goal of this study was to investigate phenotypic heterogeneity within primary and between primary and metastatic breast cancer lesions, using a novel mRNA-based assay platform to measure the functional activity of relevant signal transduction pathways on multiple samples pertaining to the same patient. Within the primary tumor, heterogeneity was estimated at macro-scale and at micro-scale. Macro-scale heterogeneity analysis was performed on up to five spatially distributed PT tissue blocks from 35 primary breast cancers of various subtypes; micro-scale heterogeneity analysis was performed on four adjacent samples of a single tissue block from 17 matched PT. To investigate phenotypic heterogeneity between primary tumor and metastatic breast cancer lesions, samples from 9 PT with matched lymph node (*n* = 33) and 9 PT with distant metastatic sites (*n* = 12) were analyzed. Table 1 gives a summary description of the sample sets used in this study. A detailed description with the respective sample numbers per patient is given in Appendix A. Measured pathway activity scores for all samples are given in Appendix A.

Archival samples were retrospectively collected under appropriate Dutch ruling and analyzed in a blinded manner; all tissue samples were formalin-fixed paraffin-embedded (FFPE). Breast cancer molecular subtyping was performed on surgically resected PT in a standard manner using the surrogate definitions from the 13th St Gallen International Breast Cancer Conference [26], based on ER, PR and HER2 protein staining (Figure 1A). Thus, in this study, the nomenclature Luminal A and B should be interpreted as Luminal A-like and Luminal B-like.

### 2.2. Tissue Sample Sets

A variable number of tissue blocks, taken from different locations, was available per PT to investigate heterogeneity in pathway activity at the macro-scale. To investigate heterogeneity at the micro-scale, four adjacent “quadrant” samples from the same tissue block (one block per tumor) were made available for a subgroup of PT. To investigate variation in pathway activity between primary tumor and metastases, two separate patient sample sets were analyzed: one with multiple PT blocks matched with a variable number of LN metastasis samples, the other containing a single PT block and a variable number of metastasis samples from different DS.

### 2.3. Sample Preparation

Tissue slides were cut from the FFPE blocks at 10 μm thickness. The tumor area was annotated by an experienced pathologist and macrodissected, aiming at similar tumor tissue volumes for RNA extraction (Figure 1B,C and Appendix A). RNA was isolated from the annotated areas using standard procedures, as described in detail in the Appendix A.

### 2.4. Measuring Pathway Activity

Functional activity of signal transduction pathways was assessed using biologically validated computational pathway models as described before [19,20,21,25]. The approach uses mRNA levels of a number of a pathway’s direct target genes, which were selected based on extensive proof points from the literature, as increased levels of expression are direct evidence of activation of the respective pathways. Underlying the approach is a Bayesian computational model, as shown in Appendix A, which describes (i) how the expression of the target genes depends on the activation of the respective transcription complex, and (ii) how qPCR results depend in turn on the expression of the respective target genes. After calibration of the model parameters using samples with a known ground truth status of pathway activity, the activity of a new test sample can be assessed by feeding its mRNA measurements in the bottom nodes of the network and applying Bayesian reasoning to determine the odds that the pathway was activated or not. After a logarithmic transformation and normalization, this yields a pathway activity score on a 0–100 scale, where 0 corresponds to the lowest and 100 corresponds to the highest odds in favor of an active pathway that a specific model can theoretically infer.

The pathway activity assays were originally developed on data from Affymetrix HG-U133Plus2.0 (Thermo Fisher, Waltham, MA, USA) microarrays but have been converted to qPCR mRNA measurements as input, to enable use of FFPE material [25]. Typically, per pathway, a slightly smaller subset of about 12 target genes were used, and models were recalibrated on qPCR measurements. For details, see the Appendix A) and [23,24]. Next to pathways’ direct target genes, a panel of reference genes was used for normalization purposes.

In our study, we used qPCR pathway assays for the AR, ER, PI3K, HH, TGFβ and Wnt pathways, available on the FIPA Pathway Plate 1.0 (Philips Molecular Pathway Diagnostics, Eindhoven, The Netherlands). As such, these represent relevant hormone-driven pathways (AR and ER), growth factor pathways (PI3K), and stem cell-related pathways (HH, TGFβ and Wnt), which play a role across different breast cancer subtypes. PI3K pathway activity is based on the inverse activity of the measured FOXO transcription factor activity score, on the premise that no cellular oxidative stress is present; for this reason, the FOXO activity score is interpreted in combination with SOD2 target gene expression level to distinguish between growth control- and oxidative stress-induced FOXO activity, as described before [20,27].

### 2.5. Statistical Data Analysis

Due to the heterogeneous sample set, we used linear mixed models [28,29] to enable the best possible quantification and statistical underpinning of the pathway analysis results. In view of readability, only a brief overview of the statistical approach is given here; for an extensive description, including detailed analysis results, we refer to the Appendix A. The model used to analyze heterogeneity in primary breast cancer subtypes considers patients grouped by cancer subtype classification, with multiple tumor block measurements per patient. The pathway activity score of a given pathway is modelled as (Figure 2A): *y_pq_ = μ_t_ + α_p_ + β_pq_.*(1)

Here, *y_pq_* is the pathway activity score of patient *p* for tumor block *q*; *μ_t_* is the average of scores in this subtype classification group *t* (e.g., LumA or LumB); *α_p_* is a random contribution of the patient tumor as a whole, compared to the group average (assumed normally distributed around zero with standard deviation *σ*_pat_); and *β_pq_* is the random contribution of block *q* which is used to accommodate tumor heterogeneity for patient *p* (assumed normally distributed around zero with standard deviation *σ*_block*,t*_ ). The value *σ*_block*,t*_ describes the spread of scores within a single patient tumor (the tumor heterogeneity) for patients in a specific breast cancer subtype *t* and *σ*_pat_ describes the variation between individual patients, regardless of the breast cancer subtype they belong to.

The measured pathway activity scores, in combination with this statistical model structure, are used for the statistical estimation of the parameters like *σ*_block*,t*_ or *σ*_pat_. Since they are fitted parameters to the data, the models provide a confidence interval for these quantities. For the analysis performed with quadrant measurements, the model was extended to accommodate and estimate additional micro-scale heterogeneity. The model used has an additional layer that enables taking both the tissue block and the quadrant sample measurements into account (Figure 2B):*y_pqe_ = μ_t_ + α_p_ + β_pq_ + ε_pqe_.*(2)

Macro-scale heterogeneity (from blocks) and micro-scale heterogeneity (from quadrants) are defined in terms of parameters of this model. Analysis was performed in STATA (StataCorp, College Station, TX, USA, 2017, Release 15 [28]) and R (www.r-project.org, accessed on 18 December 2020) using nlme and emmeans packages [29,30,31]. For details, see Appendix A.

## 3. Results

### 3.1. Signal Transduction Pathway Activity in Primary Breast Cancer Subtypes

To investigate heterogeneity in the activities of the ER, AR, PI3K-FOXO, HH, TGFβ, and WNT signaling pathways, we analyzed samples from up to five spatially distributed PT tissue blocks from 35 patients with breast cancers of various subtypes (Table 1 sample sets I and II). For this analysis, pathway activity scores were available from Luminal A, B, and ER-negative (ER−) tumors; see Figure 1 and the Materials and Methods section for a description of the subtype classification and tissue sample sets. Mean pathway activity score values per patient categorized by breast cancer subtype are presented in Figure 3A. An overview of all measured pathway activity scores is given in Appendix A. Figure 3A shows that the ER pathway had the broadest range in activity scores and largest separation in scores between subtypes. ER pathway activity scores were similar between Luminal A and Luminal B PT, but markedly lower in ER− patients. FOXO pathway activity scores (as an inverse readout for PI3K pathway activity) and TGFβ pathway activity scores were higher in Luminal A and progressively lower in Luminal B and ER− cancers, indicating the lowest PI3K pathway activity in Luminal A tumors. WNT pathway activity scores were higher in ER− cancers compared to Luminal type (Figure 3A, Appendix A).

### 3.2. Variance Explained per Model Parameter

Subsequently, we estimated the statistical variation in pathway activity between patients and within the same tumor (Figure 3B,C) using the model illustrated in Figure 2A and Appendix A). For all signaling pathways, the variation in pathway activity within a single tumor for patients in a specific subtype (*σ*_block*,t*_, for *t* = LumA, LumB, or ER−) was comparable for all breast cancer subtypes (Figure 3B, Appendix A). This implies that there is no need to compute a separate standard deviation (*σ*_block*,t*_) for each subtype. Instead, a simpler model could be used in our subsequent analysis, in which the estimate for the spread of scores within a single tumor is the same regardless of subtype (i.e., replacing *σ*_block*,t*_ by *σ*_block_). Further analysis, using this simpler model (Appendix A), indicated that the variance in pathway activity score between patients (*σ*_pat_) is larger than the variance within a single tumor (*σ*_block_) for the WNT and ER pathways (i.e., *σ*_pat_/*σ*_block_ > 1), while both variances were comparable for the remaining pathways (Figure 3C, Appendix A).

### 3.3. Variation in Pathway Activity within a Single Tumor in Primary Breast Cancer

To compare the variation of scores at macro-scale to the variation at micro-scale, pathway activity scores measured in the four quadrant samples obtained from a single FFPE block (micro-scale), as well as in all tissue blocks (macro-scale), were analyzed (sample set II, 17 patients, see Table 1 and Figure 4). Figure 4A illustrates the range and variation of pathway activity scores measured in all tissue blocks and quadrant samples of sample sets I and II (Table 1). Figure 4B depicts the spread between pathway activity scores measured in the 4 quadrant samples (micro-scale) versus the spread in scores measured in the multiple tissue blocks of the same tumor (macro-scale). Figure 4C correlates the averaged value for pathway activity of the quadrant samples and the respective matched tissue block sample. While variation in pathway activity was observed between quadrant samples and between tissue blocks of the same tumor (Figure 4B), the averaged pathway activity scores of the quadrants strongly correlated with both the score of the matched tissue block and with the average scores of the tissue blocks (Appendix A).

To determine whether pathway activity varied more at macro-scale (tissue blocks) or at micro-scale (block quadrants), we calculated the ratio between the standard deviation of pathway activity score at the macro-scale and the standard deviation of pathway activity score at the micro-scale for each signaling pathway. The standard deviations were estimated using the model illustrated in Figure 2B (details in Appendix A). For the AR, HH, TGFβ and Wnt pathways, variation in pathway activity score was significantly higher at the micro-scale than at the macro-scale, with the highest statistical significance for the HH pathway, *p* < 0.001 (Figure 4D, Appendix A). For the ER and FOXO pathways, the variation in score at the macro-scale was comparable to the micro-scale. Summarizing the macro (block level) and micro (quadrant level) heterogeneity per patient by taking the square root of the average variance across the six pathways also shows a higher variation at the micro-scale than macro-scale (Figure 4E). The median value across patients is 3.2 and 3.8 at the block level and quadrant level, respectively.

Extensive noise analysis was performed to identify technical noise, which might have biased the pathway activity scores, and therefore, the ratio between variation at macro-scale and micro-scale (Appendix A, technical noise analysis, Appendix A). High qPCR Cq values, which were more frequent in measurements of very small samples, may be associated with increased technical noise. Removal of such samples resulted in loss of significance of the higher variation in AR and Wnt pathway activity at micro-scale but did not change the results for the HH and TGFβ pathways (Figure 4D bottom). Taking this into account, we can safely draw an overall conclusion that heterogeneity in signaling pathway activity was not larger at the macro-scale (blocks) than at the micro-scale (quadrants), except for the PI3K-FOXO pathway for which a higher variation at macro-scale could not be excluded, based on the 95% confidence interval computed when the high qPCR Cq values had been removed (Figure 4D bottom). Variation in ER pathway activity was similar at micro-scale and macro-scale and variation in HH and TGFβ pathway activity was higher at micro-scale.

### 3.4. Variation in Signaling Pathway Activity at Micro-Scale versus Macro-Scale

Since the observed higher variation in HH and TGFβ pathway activity on micro-scale compared to macro-scale seemed counter-intuitive, a hypothetical computational checkerboard model was developed to help explain observed results (Figure 4F, Appendix A). In this checkerboard model, squares represent cancer cell clones with variable pathway activity scores, depicted in grey and white, which are present across a tumor. In case cancer clones are smaller than the tissue block and around the size of the quadrant samples (Figure 4F, left), the pathway activity scores measured in (all) the tissue block samples will average out the varying pathway activity scores measured in the quadrant samples, causing the variance between tissue blocks to be smaller than the variance between quadrants. This is likely to be the case for the HH and TGFβ pathways for which the activity scores were found to dominantly vary between the quadrants of a block. Variation in ER pathway activity was relatively small and more or less similar across the whole PT. This can be explained by either a homogeneous tumor, or variations in ER pathway activity at a very small scale (very small “clones” with varying ER pathway activity; Figure 4F middle) across the whole tumor. In this case, the quadrant samples are larger than the clones with varying ER pathway activity and the differences in score between quadrants and between tissue block samples are similar. Finally, clones that are larger than the sampled blocks and quadrants but much smaller than the whole tumor (Figure 4F, right) could explain the potentially higher variation at the macro-scale in PI3K-FOXO pathway activity. A detailed explanation is provided in the Appendix A.

### 3.5. Differences in Pathway Activity between Primary Tumors and Matched Lymph Node Metastases

To examine whether pathway activity in LN metastasis is similar to pathway activity in PT, multiple PT and LN matched samples per patient were analyzed (nine patients, PT tissue blocks from a subset of sample sets II and III, LN samples from sample set IV, see Table 1 and Appendix A). For all signaling pathways, activity scores within the PT were positively correlated with pathway activity scores in corresponding LN metastases (Figure 5A). However, correlations were much smaller than the correlations obtained when comparing PT block and matched quadrant samples, ranging from 0.448, 95% CI (−0.308, 0.857), *p* = 0.2, for the TGFβ pathway to 0.825, 95% CI (0.356, 0.962), *p* = 0.006, for the HH pathway (Appendix A). This indicates that pathway activity measured in PT may not reflect well the pathway activity measured in the corresponding LN metastasis.

Quantitative differences in pathway activity score between LN metastatic samples and PT samples were calculated by comparing, for each patient *p* with matched PT and LN metastatic samples, (a) the delta in pathway activity score between each lymph node metastasis and the average score of the respective PT samples, *Δ_p,_*_LN_, and (b) the delta in score between each PT tissue block and the respective PT block averages*, Δ_p,_*_block_ (Appendix A). For all pathways, the spread in *Δ_p,_*_LN_ was larger than the spread in *Δ_p,_*_block_, indicating that difference between PT and LN metastases was larger than the variation in pathway activity within the PT (Appendix A). For the TGFβ and FOXO pathways, activity scores were generally lower in lymph node metastases compared to the PT, with lower FOXO indicating higher PI3K pathway activity; for AR, HH and WNT pathways, activity scores were more frequently higher in lymph node metastases compared to the PT (Figure 5A, Appendix A).

### 3.6. Differences in Pathway Activity between Primary Tumors and Matched Distant Metastases

For this analysis, a single PT sample (per patient, *n* = 9) matched with a variable number of DS metastatic samples was available (sample group III). Except for the TGFβ pathway, correlations between DS metastasis and matched PT were worse than correlations between LN metastasis and matched PT. Correlations remained positive for FOXO and the HH and TGFβ pathways, but became negligible for the AR and WNT pathways, and negative for the ER pathway, ranging from positive correlation of 0.601, 95% CI (−0.105, 0.904), *p* = 0.09 for the HH pathway to a negative correlation of −0.359, 95% CI (−0.826, 0.401), *p* = 0.3, for the ER pathway (Figure 5B, Appendix A).

To further examine the differences in pathway activity score between distant metastases and PTs, for each patient *q* with matched PT and DS metastatic samples, deltas in activity score between each DS metastatic sample and the one available matched PT sample, *Δ_q,_*_meta_, were compared to the previously computed PT tissue block deltas*, Δ_p,_*_block_ (Appendix A, Appendix A, the indices *q* and *p* emphasize the use of two different patient cohorts). Here again, the spread in *Δ_q,_*_meta_ was larger than the spread in *Δ_p,_*_block_ for all pathways, indicating that, as with LN metastasis, differences in pathway activity scores between PT and DS metastases are larger than the variation within the PT (Appendix A). While a loss of AR, ER and TGFβ pathway activity was frequently observed in metastases compared with the PT, WNT pathway activity was regularly increased in metastases (Figure 5B, Appendix A).

### 3.7. Comparing Pathway Activity between Lymph Node and Distant Metastases 

Variation between distant metastases and lymph node metastases were analyzed by comparing the spread in *Δ_q,_*_meta_ to the spread in *Δ_p,_*_LN_. Apart from the TGFβ pathway, variation between distant metastases (spread in *Δ_q,_*_meta_) was larger than variation between lymph nodes (the spread in *Δ_p,_*_LN_, Appendix A).

### 3.8. Pathway Activity Related to Metastatic Organ Site

Specific tissues are thought to provide a favorable niche for metastatic growth that is driven by a signaling pathway for which the activating ligand is provided by the tissue niche [32]. The small number of metastatic samples from a similar organ site precluded investigation of a relation between signaling pathway activity and metastatic location. Given this limitation, pathway activity scores in individual metastatic tumors may still provide interesting information. Wnt pathway activity score was highest in bone (*n* = 2) and ileum (*n* = 1); HH pathway activity was highest in ovarian, ER pathway activity highest in a brain, and TGFβ highest in a skin and a bone metastatic lesion (see Appendix A).

### 3.9. PI3K-FOXO Pathway Analysis

For only three patients, all triple negative subtype, FOXO activity was high in combination with elevated SOD2 expression level, indicating oxidative stress-associated FOXO activity, precluding direct inference of PI3K pathway activity (Appendix A, [20]). For all other analyzed samples, PI3K pathway activity could be directly (inversely) inferred from the respective FOXO activity score.

## 4. Discussion

To investigate variation in tumor-driving signaling pathway activity within primary breast cancer tumors and between matched primary tumor and LN and DS metastases, activity of the PI3K growth factor pathway, the ER and AR hormonal pathways, and the developmental Wnt, HH and TGFβ pathways was measured at multiple locations in PT as well as in metastatic samples. Pathway activity scores were quantified using a novel method for signaling pathway analysis, adapted for FFPE samples [19,20,21]. While initial semi-quantitative analysis of the various groups of pathway measurement results already indicated our major findings [33], the complex relationships between the available sample sets required an innovative statistical model-based data analysis approach to objectively quantify our results [29].

### 4.1. Variation in Pathway Activity between Breast Cancer Subtypes

As expected, based on previous work, ER pathway activity was mostly defined by the breast cancer subtype and scores were much higher in PT of Luminal A and B patients, compared to ER− tumors [19,23]. More interesting is that within Luminal A and B subtypes, activity of the ER pathway showed a large dynamic range. We previously showed that ER protein expression is a prerequisite, but not sufficient, for functional activity of the ER pathway, and the current results further confirm this [19]. In the absence of activating mutations in the ER gene, the presence of ER ligands and specific downstream proteins (e.g., cofactors) determine the actual activity of the ER transcription factor. This creates the necessity to measure the functional activity of the ER pathway to optimally predict sensitivity to hormonal therapy [23,34]. The highest activity scores of the FOXO transcription factor, which is associated with an inactive PI3K pathway, were found in Luminal A cancers. This is in agreement with our earlier findings and with the known tumor-driving role of the ER pathway, in the absence of PI3K pathway activity, in this subtype [20].

### 4.2. Within Tumor Heterogeneity of Signaling Pathway Activity

To the best of our knowledge, to date, no information is available on phenotypic heterogeneity within primary breast cancer with respect to functional signaling pathway activity. Knowledge on signaling pathway heterogeneity is limited to studies on variations in ER and HER2 IHC staining, and on spatial variations observed in specific gene mutations and copy number changes across the tumor [17,18,35,36,37,38]. However, this does not provide information on variations in functional signaling pathway activity. In the current study, with a possible exception of the PI3K pathway, such pathway activity variation was found be comparable both at the macro-scale, across the primary tumor, and at the micro-scale, within a single tumor tissue block.

The HH and TGFβ pathway showed consistently higher variation in activity at the micro-scale. Both pathways are typically active in stem cells, suggesting that multiple small clones may be present and spread throughout the PT and composed of cancer stem cells. This fits well with the cancer stem cell hypothesis, which assumes that small groups of relatively quiescent cancer cells that have acquired stem cell characteristics are present in tumor tissue [1,39,40]. Both pathways are generally thought to play an important role in governing metastatic behavior and therapy resistance of cancer stem cells, another hallmark of cancer [1]. Previous pathway analysis, using the same pathway activity measurement approach, suggested that overt HH pathway activity, measured in a standard tissue slide, was associated with a worse prognosis [41]. In the current study, no outcome data were available, preventing further investigation of the prognostic role of activity of this signaling pathway at the micro-scale.

For the ER pathway, variation in pathway activity within the PT was lower than for the other pathways, and similar at the micro-scale versus macro-scale. The variation within the primary tumor was also relatively small compared to the variation in ER pathway activity between individual luminal A/B patients. These observations can be readily explained by considering the ER pathway as the prime tumor-driving pathway in luminal breast cancer, in contrast to HER2 and triple negative cancer subtypes. Variations in ER immunohistochemistry (IHC) staining within an ER positive primary tumor have been reported, but not related to pathway activity [36]. ER-activating estrogens are expected to distribute relatively evenly in tissue, which is compatible with only small differences in pathway activity as we observed before [35,42,43].

### 4.3. A “Checkerboard” Clone-Size Cancer Model and “Big Bang” Type of Cancer Evolution

Our observation that, with the possible exception of the PI3K-FOXO pathway, the variation in pathway activity was generally not larger at the macro-scale than micro-scale, and for the HH and TGFβ pathways, even higher at the micro-scale, is compatible with currently emerging ideas on a Big Bang type of cancer evolution based on genomic analysis of PT, in which macro-scale heterogeneity appeared to be not the dominant form [17]. Compatible with this model, the presented “checkerboard” model for cancer heterogeneity can be used to explain our findings in terms of variable cancer cell clone sizes, all spread more or less evenly across the tumor: the smallest “clones” or cancer cell areas with slightly varying ER pathway activity; small clones that have an active HH and/or TGFβ pathway, and possibly intermediate/large clones with varying PI3K pathway activity. In such a model, combined ER and PI3K pathway active clones may be more rapidly proliferating, while HH and TGFβ pathway active clones are “neutral”, less proliferative, and may have a stem cell character.

### 4.4. Variation in Pathway Activity between Primary and Lymph Node Metastases

Subsequently, we investigated whether pathway analysis of the primary tumor could predict pathway activity in metastatic tumors. While pathway activity scores between quadrant samples of a single block, and between blocks of one PT showed highly significant correlations, correlations decreased between PT and LN metastases and significance was lost. The lowest correlation was found between PT and DS metastases. These results provide interesting new information about breast cancer metastasis. Since LN metastatic samples were spatiotemporally close to the PT, a closer relation to the PT was expected compared to the DS metastases that are spatiotemporally further away. The PI3K pathway is generally thought to be an important metastasis-driving pathway in addition to its general role as a tumor “survival” pathway, explaining the relatively strong correlation in activity between PT and LN metastases [44]. The observed higher variation in pathway activity between DS metastases compared to LN metastases is probably due to LN representing a singular type of metastatic niche, while DS metastases grow in all kinds of organ niches with their respective variations in the microenvironment, including ligand availability for the various signaling pathways [45,46]. As a few tentative examples to illustrate this, Wnt ligand availability may have caused the one intestinal metastatic tumor to have the highest Wnt pathway activity score, while local Sonic Hedgehog ligand availability may have induced the high HH pathway activity score in the ovarian metastatic lesion [32,47,48]. For the ER pathway, such heterogeneity has also been demonstrated in ER-positive breast cancer patients [49].

Interestingly, a negative correlation was present for ER pathway activity between PT and DS metastases. Since only ER-positive luminal patients were included in this part of the study, a number of patients probably had received adjuvant hormonal therapy, potentially resulting in selection for ER pathway-inactive metastases [50]. Unfortunately, lack of treatment information precluded further analysis of this relationship between treatment and ER pathway activity.

Obvious limitations of the study were the relatively limited patient and sample numbers, and lack of clinical treatment and outcome data. Despite this, to the best of our knowledge, the study is unique in allowing a quite detailed comparison between signaling pathway activity scores within a PT and between PT and metastases.

## 5. Conclusions

The clinically relevant conclusions that can be drawn suggest that heterogeneity with respect to pathway activity within a single biopsy may generally be representative for the whole tumor, with the possible exception of the PI3K pathway, while variation in ER pathway activity is relatively small within the PT compared to that seen between luminal breast cancers from different patients. We cannot exclude that for the PI3K pathway, multiple spatially separated biopsies may be required to obtain a sufficiently complete picture of this pathway’s activity pattern across the tumor. Pathway activity scores found in the PT predicted activity in LN metastases to some extent, especially with respect to the PI3K-FOXO pathway. Prediction of pathway activity in DS metastases is no longer realistic, also in view of the large variation between metastases in different organ locations. If possible, taking biopsies from multiple metastatic locations may, therefore, be preferred when considering targeted therapy for metastatic disease, to choose the most effective (targeted) drug or drug combination.

In view of its high clinical relevance, especially with respect to targeted treatment in (neo)adjuvant and metastatic settings, the current findings should be confirmed in subsequent clinical studies. We recommend that a confirmation study should: (1) optimize standardization of tissue sampling; (2) note size of the tumor and location of the sample taken in the 3D space of the tumor, to reduce the influence of variable microenvironment, e.g., oxygen concentration, immune cell infiltrate, etc.; (3) use similar amounts of cancer cells to extract RNA from. Importantly, application and further development of the statistical models that we introduced should help to improve quantification of cancer heterogeneity, necessary to objectively compare clinical studies and bring the field forward. We believe that such studies will further complement knowledge on tumor evolution and the resulting tumor heterogeneity, enabling improvement in choosing the most effective therapy for patients with breast cancer.

## Figures and Tables

**Figure 1 cancers-13-01345-f001:**
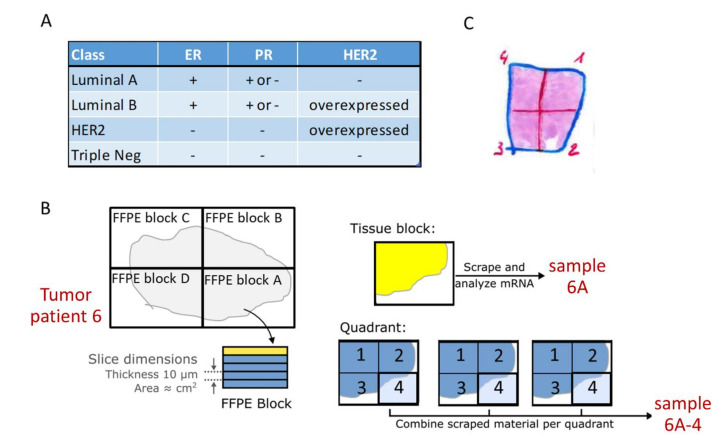
Analyzed primary breast cancer samples. (**A**) Pathology classification of analyzed primary breast cancers. Surrogate classification of Luminal A and Luminal B subtypes based on the surrogate definitions from [26], using ER/PR/HER2 immunohistochemistry staining. (**B**) Tissue samples analyzed from surgically resected primary tumors were either a tissue block sample or a quadrant sample. Tissue block samples were obtained by scraping one or more adjacent slides (depending on the amount of cancer tissue per slide) from each available FFPE tissue block. The tumor tissue area was annotated by a pathologist and tumor-containing areas were scraped from the slide(s) for RNA isolation. Quadrant samples were obtained from one randomly selected FFPE tissue block per patient. Sequential tissue slides were divided into 4 quarts, tumor areas annotated and tissue from the quarts scraped from the slides for RNA isolation. Care was taken to scrape only cancer tissue. To obtain similar amounts of tumor tissue, multiple adjacent slides were scraped until the same area of scraped tumor had been collected. (**C**) Typical example of a tissue block sample, showing how quadrant sample areas were divided. More examples are given in Appendix A.

**Figure 2 cancers-13-01345-f002:**
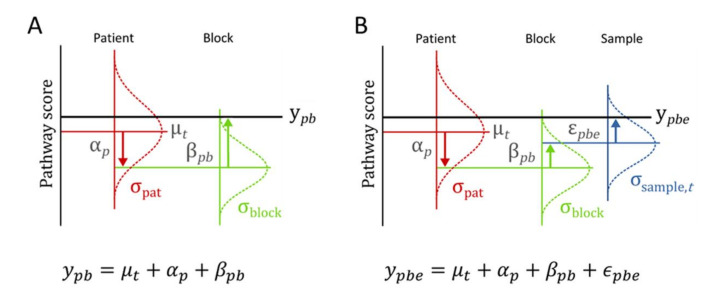
Schematic representation of models used for data analysis. (**A**) Model used for estimation of variation in signaling pathway activity between breast cancer subtypes *(y_pq_ = μ_t_ + α_p_ + β_pq_)*. (**B**) Extended model enabling additional comparison between variation at micro-scale and macro-scale. *(y_pqe_ = μ_t_ + α_p_ + β_pq_ + ε_pqe_)*. The models have random contributions that are added to each other; the random character of each level of contribution is indicated by a Gaussian distribution with standard deviation (σ) and example realizations from that distribution. The models describe the pathway activity scores *y_pb_* and *y_pbe_* as a sum of contributions. Starting from the average score *μ_t_* (computed for each breast cancer subtype *t* separately, with *t* being Luminal A, Luminal B, or ER negative), a patient-specific contribution *α_p_* (red arrows) and a block-specific contribution *β_pb_* (green arrows) are added to the average signaling pathway activity score (*μ_t_*). In the extended model, an additional contribution *ε_pbe_* (blue arrow) is added to the pathway activity score of each block, to model the possibility of quadrant-to-quadrant heterogeneity (see Figure 1B for block and quadrant definitions).

**Figure 3 cancers-13-01345-f003:**
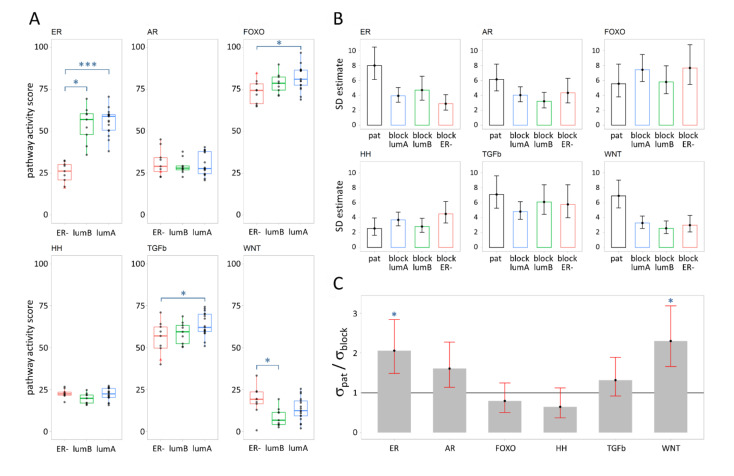
Variation in signaling pathway activity score per breast cancer subtype in the tissue block samples of sample sets I and II. (**A**) Distribution of scores for Luminal A (lumA), Luminal B (lumB), and HER2-positive/triple negative breast cancer (ER−). Each point represents the average pathway activity score across all tissue block samples of a patient. The HER2-driven patient sample is depicted as a small red triangle. Tukey adjusted significance levels of pairwise test for equality of means are indicated (***: *p* < 0.0001, *: *p* < 0.05, values in Appendix A). (**B**) Estimated standard deviation (SD) with corresponding 95% confidence intervals (95% CI) of signaling pathway activity scores describing the heterogeneity in activity scores between all patients, irrespective of subtype (pat) and within a single tumor for tumors in a specific breast cancer subtype (block lumA, block lumB, block ER−), based on pathway activity scores of primary tumor tissue block samples (values in Appendix A). (**C**) Estimated ratios of between patient SD to within tumor SD (σ_pat_/σ_block_) with corresponding 95% confidence intervals for subtype-independent *σ*_block_ model. Significance levels are indicated (*: *p* < 0.05, values in Appendix A).

**Figure 4 cancers-13-01345-f004:**
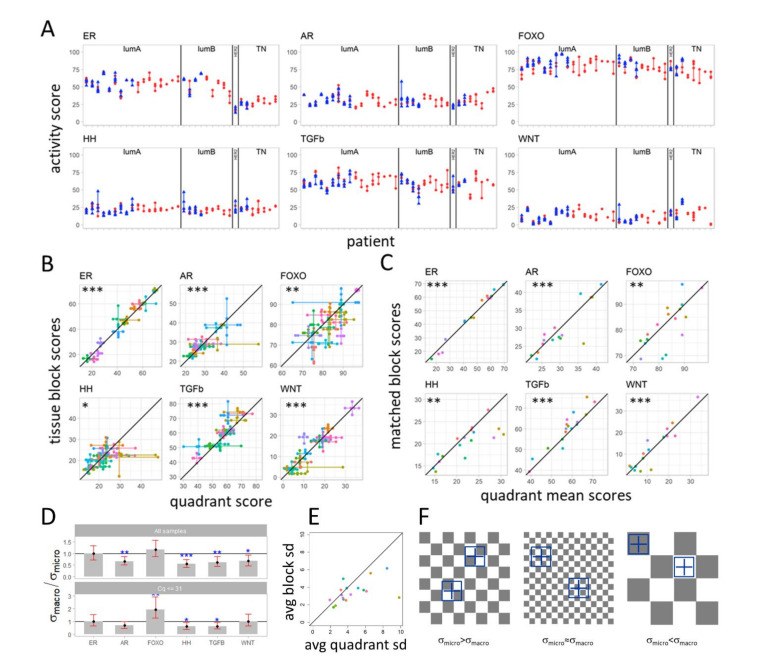
Heterogeneity in signaling pathway activity score within primary tumors at micro- and macro-scale (sample sets I and II). (**A**) Pathway activity scores for tissue block (blue) and quadrant (red) samples of each patient. Ranges are presented as vertical lines; individual scores are presented as triangles. (**B**) Spread in signaling pathway activity scores from quadrant samples (horizontal lines) vs. spread in scores from tissue block samples of the same patient (vertical lines). Significance level for the correlation between mean scores of tissue block samples versus mean scores of quadrant samples (point where horizontal and vertical lines cross) are indicated in the figure by stars (*: *p* < 0.05, **: *p* < 0.01, ***: *p* < 0.001, corresponding values are in Appendix A). (**C**) Correlation between mean quadrant sample pathway activity score and score of the corresponding tissue block sample. Stars indicate the significance level of the correlations (corresponding *p* values are in Appendix A). (**B**,**C**) Diagonal black lines illustrate a one-to-one relation. Samples of the same patient have the same color. Black lines illustrate a one-to-one relation. (**D**) Ratio between macro-scale and micro-scale standard deviation (SD) of signaling pathway activity scores; σ_macro_/σ_micro_ measured in tissue block samples and quadrant samples, respectively. Variances computed using the macro- vs. micro-scale model (Figure 2B, Appendix A) using all samples (top) or those with less technical noise (leaving out all samples with average Cq values > 31 for the reference genes used for the qPCR measurements). Significance level of Wald test *p*-value for comparing the ratio σ_macro_/σ_micro_ to 1 are indicated by stars (*p* values for model run using all samples in Appendix A). (**E**) Summarized macro (block level) and micro-scale (quadrant level) heterogeneity per patient by taking the square root of the average variance across the six pathways, where an average SD of 10 means an average confidence interval of about 40 points. (**F**) Checkerboard visualization of a primary cancer, explaining differences in heterogeneity between micro-scale and macro-scale measurements of pathway activity. Left: Squares represent (small) cancer cell clones with variable pathway activity scores, simulated by grey (high pathway activity) and white (low pathway activity). For example, analyzing HH pathway activity in a randomly localized “tissue” sample (analogous to the tissue block samples) results in a smoothed averaged pathway activity, due to canceling out of variations in pathway activity that are present in areas smaller than the sampled area. On the other hand, when taking four quadrant samples, the varying pathway activity scores in the quadrants are measured, resulting in a higher measured pathway activity heterogeneity at micro-scale. Right: Quadrangles represent large cancer cell clones, with variations in pathway activity scores. In this case, it is expected that more heterogeneity will be found at the macro-scale, since the quadrants are more likely to have the same pathway activity. This might be the case for the PI3K-FOXO pathway. For detailed information on the associated statistical model, see Appendix A.

**Figure 5 cancers-13-01345-f005:**
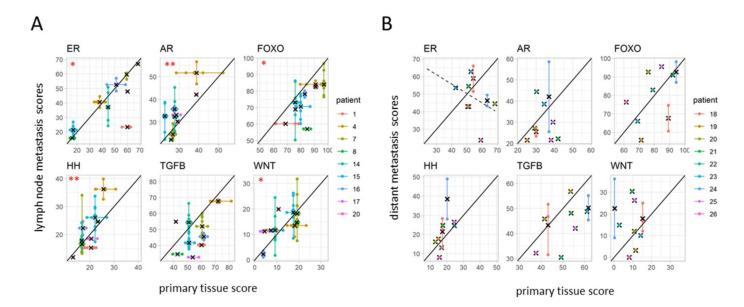
Heterogeneity in signaling pathway activity between primary tumor and metastases (**A**) Correlation between measured pathway activity scores in primary (x-axis) tumor and matched lymph node metastases (y-axis) visualized by plotting pathway activity scores (Sample sets IV and II/III). (**B**) Correlation between measured pathway activity score in primary tumor (single sample) and matched distant metastases (Sample set III). Per plot, colors indicate samples that belong to one patient (patient IDs indicated in the legends), the cross indicates the average, and the black line is a visual guide for a one-to-one relation. Significance level for the correlation between mean scores are indicated by stars (*: *p* < 0.05, **: *p* < 0.01, corresponding values are in Appendix A).

**Table 1 cancers-13-01345-t001:** Summary description of sample sets used. The total number of patients and number of patients per subtype are given. Between brackets are the corresponding number of samples and their type; b: Primary tumor tissue block sample, PT: primary tumor sample, LN: lymph node metastasis sample, DS: distant site metastasis sample. Detailed description is given in the Appendix A.

Sample Set	Description	Total	Per Breast Cancer Subtype			
			LumA	LumB	HER2	TN
I	1 to 5 block (b) samples from primary tumors	18 (49 b)	8 (20 b)	5 (15 b)	-	5 (14 b)
II	2 to 5 block samples from primary tumors with 4 matched quadrant samples (per patient)	17 (50 b)	9 (28 b)	4 (12 b)	1 (2 b)	3 (8 b)
III	Primary tumors (PT) and 1 to 3 matched distant metastasis samples	10 (9 PT/13 DS)	Subtyping not available			
IV	1 to 10 lymph node metastasis from 9 patients from sample sets II and III	9 (33 LN) *	4 (20 LN)	2 (3 LN)	1 (1 LN)	1 (8 LN)

* Subtyping not available from one patient.

## Data Availability

Pathway activity scores and associated info used in the analyses are given in Appendix A.

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
