# Peer review of "Heterogeneity in Signaling Pathway Activity within Primary and between Primary and Metastatic Breast Cancer"

_cancers, 2021, doi:10.3390/cancers13061345_

Round 1

Reviewer 1 Report

All comments have been addressed. 

Reviewer 2 Report

The authors have addressed my concerns and substantially improved the manuscript.

This manuscript is a resubmission of an earlier submission. The following is a list of the peer review reports and author responses from that submission.

Round 1

Reviewer 1 Report

The manuscript present an interesting approach of analyzing pathways in subcompartment of tumors and in metastasis. This has been investigated at the mutational level, but not so much at gene expression and pathway level. Many different biopsies have been obtained from the patient samples which is also a quality of the study.

However, there are some severe caveats of the study: 

  • FFPE tissue is in general problematic especially for RNA analyses. It seems like the authors obtain quit clear results for estrogen receptor (ER), whereas the other genes are noisy and not really different between subgroups.
  • The statistical methods are very well described and seems appropriate. However, the laboratory method is only in supplementary information and not that clearly described. How many genes are measured? Only 6?
  • Molecular subtypes are identified using few genes. It is questionable if this can be used as a valid proxy for Sørlie or PAM50 subtypes. Was HER2 measured? How should HER2 subtype be measured without this gene? How to separate lumA and lumB?
  • A general problem in analyzing gene expression and not mutations in cancer heterogeneity studies is that the content of normal tissue is likely to vary. The authors claim that care was taken to scrape only cancer tissue. However, this is not really documented.
  • The contamination with normal tissue is in particular a challenge in metastasis where the normal tissue is from another organ. This will strongly bias gene expression data.
  • The figures have poor quality. For example part of figure 3 is missing and the text below bars is not readable.
  • Figure numbering is completely wrong. Figure 3 with the main results is referred to as figure 1 in most places.

Reviewer 2 Report

In the manuscript entitled "Heterogeneity in signaling pathway activity within primary and between primary and metastatic breast cancer", the authors aimed to investigate the heterogeneity with primary tumor, and between primary and metastatic tumors in breast cancer. The authors used a qPCR-based platform and established computational models to define the heterogeneity of primary tumor, and the heterogeneity between primary tissue and metastatic tumors. Through the study, they found heterogeneity in primary tumor was not larger at the macro-scale  than at the micro-scale except for the PI3K-FOXO pathway. The correlations between DS metastasis and matched PT were weak, indicating it is more heterogeous bewteen primary tumor and metastatic tumors. Overall, the concept of this study is interesting and is of significance in the field. Regarding the current manuscript, I have some concerns: 

1) The authors only measured the activity of androgen receptor, estrogen receptor (ER), PI3K-FOXO, Hedgehog, TGFβ, and WNT signaling pathways to indicated the heterogeneity. However, both the tumor cells and tumor microenvironment are complex, so only choosing those pathways for the study is biased. If those signaling pathways are not the predominent pathways under some conditions, they can even not be used to represent the heterogeneity. It is better to define the heterogeneity using more signaling pathways and a larger scale even not genome scale.

2) The authors checked the correlation of those signaling pathways within PT and between PT and metastasis tumors to reflect the heterogeneity. Can you define a more direct parameter based on the models to indicated heterogeneity? e.g. a parameter from 0-1, 0 indicates no heterogeneity and 1 indicates heterogeneous samples. In this case we can more easily get an idea how heterogeneous the sampels/tumors are.

3) Even for different signaling pathways, I think the weight / contribution to the heterogeneity is different. How did you determine the weight/contribution of each variance to the heterogeneity?

4) The quality of the figures is not good, especially Figure 3. I can hardly see the labels in the figures, and even part of the figure 3B and 3C is missing. 

5) I think the overall conclusion from this study is somehow close to what I thought. Escpecially if the author can provide more information how different it is between primary tissue and and metastatic tumors, it will be much better. In that way, it will help us understand how the primary tumor cells are transformed into metastatic cells.

6) What is your reference / internal control gene for qPCR? and can you list the genes and primers you used to define those signal pathways?